# The Mechanisms of BDNF Promoting the Proliferation of Porcine Follicular Granulosa Cells: Role of miR-127 and Involvement of the MAPK-ERK1/2 Pathway

**DOI:** 10.3390/ani13061115

**Published:** 2023-03-21

**Authors:** Xue Zheng, Lu Chen, Tong Chen, Maosheng Cao, Boqi Zhang, Chenfeng Yuan, Zijiao Zhao, Chunjin Li, Xu Zhou

**Affiliations:** 1Laboratory for Regulation of Reproduction, College of Animal Sciences, Jilin University, Changchun 130062, China; 2College of Biological and Pharmaceutical Engineering, Jilin Agricultural Science and Technology University, Jilin 132101, China

**Keywords:** BDNF, porcine, proliferation, microRNA, CCND1, ERK

## Abstract

**Simple Summary:**

Recently, increasing the efficiency of porcine embryo cultures by promoting oocyte maturation in vitro has attracted much attention. Brain-derived neurotrophic factor (BDNF) was beneficial to oocyte maturation and increased the developmental potential of porcine embryos. Although the effects of BDNF on porcine follicular development and the maturation of oocyte have been previously demonstrated, no literature was available, at the time of this work, relating to miRNA-regulated gene expression and signal pathways in mechanisms of BDNF, promoting porcine GCs proliferation. Therefore, this study explored the miRNAs involved in BDNF-induced proliferation of porcine GCs, as well as the involvement of the MAPK-ERK signaling pathway.

**Abstract:**

As a member of the neurotrophic family, brain-derived neurotrophic factor (BDNF) provides a key link in the physiological process of mammalian ovarian follicle development, in addition to its functions in the nervous system. The emphasis of this study lay in the impact of BDNF on the proliferation of porcine follicular granulosa cells (GCs) in vitro. BDNF and tyrosine kinase B (TrkB, receptor of BDNF) were detected in porcine follicular GCs. Additionally, cell viability significantly increased during the culture of porcine GCs with BDNF (100 ng/mL) in vitro. However, BDNF knockdown in GCs decreased cell viability and S-phase cells proportion—and BDNF simultaneously regulated the expression of genes linked with cell proliferation (CCND1, p21 and Bcl2) and apoptosis (Bax). Then, the results of the receptor blocking experiment showed that BDNF promoted GC proliferation via TrkB. The high-throughput sequencing showed that BDNF also regulated the expression profiles of miRNAs in GCs. The differential expression profiles were obtained by miRNA sequencing after BDNF (100 ng/mL) treatment with GCs. The sequencing results showed that, after BDNF treatment, 72 significant differentially-expressed miRNAs were detected—5 of which were related to cell process and proliferation signaling pathways confirmed by RT-PCR. Furthermore, studies showed that BDNF promoted GCs’ proliferation by increasing the expression of CCND1, downregulating miR-127 and activating the ERK1/2 signal pathway. Moreover, BDNF indirectly activated the ERK1/2 signal pathway by downregulating miR-127. In conclusion, BDNF promoted porcine GC proliferation by increasing CCND1 expression, downregulating miR-127 and stimulating the MAPK-ERK1/2 signaling cascade.

## 1. Introduction

Ovarian follicle development is a fundamental process of reproductive physiology in female mammals. Germ stem cells differentiate into oogonia in the genital ridge and the oogonia divide and become primary oocytes via mitosis. The primordial follicles are formed during the process of differentiation. Thus, primordial follicle assembly is crucial for the acquisition of fertility during female mammalian reproduction [1]. Primordial follicles are composed of primary cells and surrounding GCs and basement membrane [2]. The proliferation of GCs can effectively promote follicle development and the formation of the antral cavity. As such, improving the effectiveness of in vitro porcine embryo production by promoting the maturation of oocytes has become a research hotspot.

Neurotrophins (NTs) are soluble polypeptides known for their effects in regulation of synaptic growth, revival, differentiation and functionalization in the nervous system [3,4,5]. The NTs not only affect nervous system development, but also participate in ovarian development [6,7]. During embryogenesis in pigs, various NTs play important roles in folliculogenesis [8,9], oocyte maturation [10,11], steroid synthesis [9] and embryonic development [11]. Brain-derived neurotrophic factor (BDNF) is a well-known member of the NT group and may also increase oocyte maturation and porcine embryonic growing potential in vitro [10,11]. In adult mammalian ovaries, BDNF expression, and its high-affinity TrkB receptor, have both been detected [12]. Moreover, the expression of BDNF differed in porcine Germinalvesic oocytes and in vitro mature MII oocytes, implying that BDNF might be involved in porcine oocyte development and maturation [13]. Previous data from our laboratory showed that BDNF promoted GC growth in cocultured bovine early embryos [14] and regulated GC proliferation and progesterone synthesis via the TrkB-AKT signaling pathway [15]. Additionally, BDNF-TrkB signaling was involved in oogenesis, follicle recruitment, germ cell survival and oocyte nuclear and cytoplasmic maturation [16,17,18,19]. Thus, BDNF was shown to regulate the reproductive function of mammals along with TrkB.

As an important epigenetic regulatory mechanism, microRNAs (miRNAs) inhibit target genes’ expressions by binding to their 3′UTR region and are related to the key regulation of many biological processes. Indeed, miRNAs play critical roles in spermatogenesis [20,21], follicle development [22,23,24], maturation of oocytes [25,26] and primary embryonic growth [26,27] during mammalian reproduction. Regulation of miRNAs with BDNF involvement has been demonstrated in the nervous system [28,29,30]. In the ovary, BDNF affects oocyte–cumulus cell interaction and granulosa cell function by regulating miRNAs. For example, miR-NA-10b inhibits goat GC proliferation by targeting BDNF [31]. Moreover, BDNF promotes the expansion of porcine cumulus–oocyte complex by downregulating miR-NA-205 [11]. miR-127, as a tumor suppressor, plays a role in inhibiting tumor cell growth and migration in various types of cancer [32,33,34,35]. In mammalian reproductive systems, miR-127 inhibits the progression of ovarian cancer by regulating MAPK4 expression [36]. In addition, miR-127 affects placental development by targeting Rtl1 [37,38,39]. Notably, miR-127 was differentially-expressed in follicular fluid of polycystic ovary syndrome (PCOS) patients, compared with normal controls [40]. Therefore, the influence and role of miR-127 in the ovarian follicle is worthy of attention.

Although the impacts of BDNF on follicular development and maturation of oocytes in pigs have been demonstrated, there has been, to the best of our knowledge, no report on miRNA-regulated gene expression and signaling pathways related to BDNF mechanisms promoting porcine GC proliferation. Therefore, this study explored the mechanishms by which miRNAs were regulated by BDNF and MAPK-ERK1/2 signaling pathways in porcine GCs proliferation.

## 2. Materials and Methods

### 2.1. Cell Isolation and Culture

Porcine ovaries, from prepubertal gilts, were gathered at a local slaughter abattoir and kept at 37 °C in sterile saline, with penicillin and streptomycin at concentrations of 100 IU/mL. The ovaries were transported within 2 h to the laboratory. GCs were isolated and collected from the follicular fluid, as previously stated [15]. Concisely, the collected ovaries were rinsed three times with water to eliminate blood stains, then placed in sterile normal saline after being washed three times with 75% ethanol for disinfection. Follicular fluid was aspirated between 3 mm and 5 mm follicles using a 10 mL sterile syringe, and about 8–10 mL was acquired from twenty porcine ovaries. The collected follicular fluid was transported into 20 mL PBS and centrifuged at 300× *g* for 1 min to eliminate the oocytes. The isolated supernatant was transferred into a 50 mL sterile centrifuge tube, followed by centrifugation at 500× *g* for 6 min to separate GCs. The trypan blue exclusion test was performed to examine the viability of GCS. Approximately 1 × 10^6^ GCs per well were seeded in 6-well plates within 2 mL culture medium (Dulbecco’s Modified Eagle Media/Nutrient Mixture F12 (DMEM/F12)), comprising of 10% fetal bovine serum (FBS) for 24 h at 37 °C with 5% CO_2_.

### 2.2. Immunofluorescence Staining

Fresh porcine follicle granulosa cells were cultured in 24-well (2 × 10^5^ cells/well) optical bottom plates for 24 hand then triple-washed with PBS (5 min each). Cells were preserved for 20 min in precooled absolute methanol and then permeabilized with 0.1% Triton X-100 for 30 min. Cells were blocked with 1% goat serum at ambient temperatures for 30 min, followed by incubation at 4 °C overnight in the following primary antibodies: anti-FSHR (BS2618, Bioworld, Nanjing, China, 1:200 for rabbit anti-mouse), anti-BDNF (BS6533, Bioworld, 1:200 for rabbit anti-mouse), or anti-TrkB (BS94070, Bioworld, 1:100 for rabbit anti-mouse). They were then rinsed three times with PBS. After that, the cells were incubated in goat anti-rabbit immunoglobulin G (IgG) combined with fluorescein (FITC) antibody (BS10950, Bioworld, 1:1000) at ambient temperatures for 1 h and triple-washed with PBS. After that, cells were stained for 10 min using PI solution (1:1000; Beyotime, Shanghai, China) and observed with a fluorescence microscope (IX71; Olympus, Tokyo, Japan). The porcine GCs were presented in two colors: green for FSHR and red for nucleus.

### 2.3. Cell Treatment

GCs were cultured in 6-well plates, with 10^6^ cells per well, until cell density reached 70% and then washed twice with PBS to eliminate the suspended cells. The culture media were replaced with serum-free DMEM/F12 for 4 h, and then cells were treated with BDNF, K252α (TrkB-specific inhibitor; 100 ng/mL; Invitrogen, Carlsbad, CA, USA) or PD98059 (MAPK suppressor; 15 μM; Beyotime). BDNF (0, 10, 20, 50, 100 and 200 ng/mL; PeproTech, London, UK)-treated GCs were used, for different times (12, 24, or 48 h). K252α (100 ng/mL) were used for 24 h, PD98059 (15 μM), for 30 min.

### 2.4. Cell Transfection

The siRNA for BDNF (si-BDNF sequence: 5′-GCGGTTCATAAGGATAGAC-3′) was synthesized from RiboBio (Guangzhou, China). Transfection of porcine GCs with 50 nM siRNA was accomplished using FuGENE^®^ HD (Roche, Basel, Switzerland) reagent for transfection and Opti-MEM medium (Gibco, Paisley, Scotland, UK), as directed by the manufacturing company. GCs were transfected with siRNA for 24 h to extract RNA and protein.

The mimics and inhibitors of miRNAs (miR-185, miR-1273, miR-7047, miR-127 and miR-532) were acquired from Genepharma (Shanghai, China). Based on the manufacturer’s guidelines, mimics of the five miRNAs or miR-127 inhibitors were transfected into the GCs using FuGENE^®^ HD transfection reagent for 24 h. The transfection concentration was 50 nM.

### 2.5. Cell Viability Assay

The capacity of GCs’ viability was evaluated by CCK-8 kit (Dojindo, Shanghai, China). Briefly, 100 μL of medium, containing 1 × 10^4^ GCs, were grown in 96-well plates. After treatment (BDNF, K252α or PD98059) or transfection (miRNAs or si-BDNF), CCK8 reagent was added into medium for 10 μL of each well. After 2 h of incubation, the absorbance was detected at 450 nm with a microplate reader (BioTek, Winooski, VT, USA).

### 2.6. RNA Extraction and Reverse Transcripton-Quantitative Polymerase Chain Reaction (RT-qPCR)

Total RNA was extracted from GCs using Trizol reagent (Invitrogen, Carlsbad, CA, USA) and reverse transcription was done using a PrimeScript RT Reagent Kit (Takara, Tokyo, Japan) based on manufacturer instructions. qRT-PCR was performed with a SYBR Green Master Mix (Takara, Tokyo, Japan) and real-time quantitative fluorescence PCR instrument (Mx3005P; Agilent, Santa Clara, CA, USA). The reaction criteria were as follows for temperature profile: denaturation at 95 °C for 15 min, then amplification for 40 cycles of 35 s at 95 °C and 40 s at 60 °C. Using GAPDH or U6 as reference genes, the relative mRNA expressions of indicated genes were calculated using 2^−ΔΔCT^ methods. All sequences of primer synthesis were completed by Comate Bioscience Co., Ltd. (Changchun, China) as shown in Appendix A.

### 2.7. Western Blotting

Cell lysis and protein collection were done using a RIPA buffer (Beyotime) containing phosphatase inhibitor cocktail A (Beyotime). The concentrations of proteins were determined using a BCA Kit (Beyotime). Then, 12% SDS-PAGE gel electrophoresis was used to separate a total of 20 μg protein in each sample and the separated proteins were transferred to polyvinylidene difluoride (PVDF) membranes (Millipore Co., Ltd., Norwood, OH, USA). The unspecific bands of membranes were blocked with 5% skim milk at ambient temperature for one hour. Subsequently, the membranes were stored at 4 °C overnight with the primary antibodies listed below: anti-BDNF (BS6533, Bioworld, 1:1000, Rabbit anti-mouse), anti-CCND1 (ab16663, 1:1000), anti-P21 (ab109520, 1:500, Rabbit anti-human), anti-BCL2 (ab182858, 1:50), anti-Bax (ab32503, 1:50), anti-ERK1/2 (ab184699, 1:100), anti-p-ERK1/2 (ab278538, 1:50) or anti-beta-actin (ab179467, 1:80). After rinsing with TBST (4 × 8 min), peroxidase (HRP), coupled with secondary antibody (1:800), was used to incubate the membranes for 1 h. The bands were visualized using a BeyoECL Plus kit (Beyotime) and tested using a chemiluminescent detector (Tanon, Shanghai, China), based on the manufacturer’s instructions. Unless otherwise noted, all antibodies were rabbit anti-mouse and purchased from Abcam (Cambridge, MA, USA). Intensity of bands were analyzed via grey scanning using the Tanon Gel Imaging System (Tanon).

### 2.8. Flow Cytometric Analysis (FACS)

FACS was used to measure the cell cycles of the porcine GCs. GCs were digested by trypsin and the collected cells were twice-rinsed with precooled PBS, then fixed and refrigerated in 70% ethanol for more than 24 h. The fixed cells were resuspended in 500 μL staining buffers containing PI (50 μg/mL) and RNaseA (1 mg/mL) after rinsing with PBS. Subsequently, after 30 min incubation in the dark, BD-LSR flow cytometry (BD Biosciences, Franklin Lakes, NJ, USA) was performed to determine cell cycle kinetics.

### 2.9. Sequencing of miRNAs

GCs were collected after treatment with BDNF (100 ng/mL) for 24 h. The library operation and sequencing experiments were performed following standard Illumina procedures. The Small RNAs Sample Pre Kit (Illumina, San Diego, CA, USA) was used to structure libraries of small RNA sequencing and Illumina HiseQ2000/2500 was performed for sequencing. A sequence length of 1 × 50 bp was collected for bioinformatics analysis.

### 2.10. Bioinformatics Analysis

The Illumina HiSeq 4000 sequencing platform (Illumina, Inc., San Diego, CA, USA) was used to sequence RNA from pig GC specimens and construct cDNA libraries. Denatured libraries were converted into single-stranded DNA molecules and sequenced over 51 cycles on Illumina HiSeq, based on the manufacturer’s directions. The analyses were performed with the help of Beijing Yuanyi Gene Technology Co., Ltd. (Beijing, China). After sequencing, Solexa CHASTITY [41] quality filtered reads were harvested as clean reads. Then, clean reads were screened for siRNA through BLAST and Rfam databases and classified siRNA were annotated or predicted as mature miRNA. The miRBase database was used for sequence alignment of mature miRNA. We calculated miRNA expression using the most readily available isoforms; miRBase was used to identify the mature miRNAs and all miRNA isoforms (5p or 3p). When miRNA profiles were differentially-expressed, comparisons between the two groups, fold changes, *p*-values and FDRs were calculated and used to identify significantly differentially-expressed miRNAs. log2 (fold change) > 1 or log2 (fold change) < −1 were selected for differentially-expressed miRNAs. The differences were statistically significant (*p* < 0.05) after R package comparison.

### 2.11. Data Analysis

Statistical analyses were performed with SPSS (Windows version 19). Data analysis between the two groups was done using the unpaired t-test. Three or more groups of data were evaluated for significant differences using ANOVA. Statistical significance was shown at * *p* < 0.05 and ** *p* < 0.01. Data were shown as mean ± standard deviation (SD).

## 3. Results

### 3.1. Effects of BDNF on the Proliferation of Porcine GCs

The effects of BDNF on GC proliferation were determined. Figure 1A revealed that BDNF increased porcine GCs’ viability in a dose-dependent manner (*p* < 0.05) and a concentration of 100 ng/mL was the most effective after 24 h. Unless otherwise noted, all BDNF described in our results were treated at 100 ng/mL for 24 h.

In addition, BDNF knockdown (Figure 1B) in GCs significantly decreased GC viability (Figure 1C). The proportion of S-phase cells in the si-BDNF group (7.55 ± 0.1%) was considerably reduced, compared with the si-NC group (9.47 ± 0.42%) (Figure 1D). The levels of proliferation-related genes’ (including CCND1, p21 and Bcl2) expression and apoptosis-related genes (Bax) in porcine GCs cells were evaluated. As shown in Figure 2E, after BDNF knockdown, the mRNA and protein expression levels of CCND1, p21 and Bcl2 were significantly reduced. On the contrary, the expression levels of Bax were considerably elevated (Figure 1E).

### 3.2. The BDNF/TrkB Pathway Affects the Proliferation of Porcine GCs

The effects of BDNF on GC proliferation after K252α treatment were assessed. The results showed that the viability of porcine GCs (Figure 2A) and the distribution of cells in S-phase (Figure 2B) were significantly decreased with K252α treatment alone. Moreover, K252α reduced BDNF-induced porcine GC viability, and the proportion of cells in S-phase, when K252a and BDNF treatment were undertaken together in GCs (Figure 2A,B).

The expression levels of genes connected with cell proliferation and apoptosis were measured. BDNF significantly increased mRNA (Figure 2C-a) and protein (Figure 2C-b,C-c) expression levels for CCND1, p21 and Bcl2. Additionally, the Bax mRNA and protein expression levels were reduced dramatically. Figure 2C shows that BDNF, in combination with K252a, weakened the effects of BDNF on the expression of the above genes in GCs.

### 3.3. BDNF Promotes GCs Proliferation through Increase of CCND1 by Downregulating miR-127

miRNA sequencing was performed. According to the sequencing results, 72 differentially-expressed miRNAs were detected, including 34 upregulated and 38 downregulated, as shown in Figure 3A-a,A-b (fold change > 2, *p* < 0.05). Additionally, 5 miRNAs (miR-185, miR-1273, miR-7047, miR-127 and miR-532) associated with cellular processes (Figure 3A-c) and cell proliferation signaling pathways (Figure 3A-d) were significantly downregulated in GCs after BDNF treatment (Figure 3B).

Subsequently, to ascertain the consequences of these miRNAs on GC proliferation, GCs were transfected with the above 5 miRNA mimics for 48 h, respectively. Overexpression of miR-185 and miR-532 inhibited GC viability, while overexpression of miR-127 significantly reduced the viability of GCs and the proportion of cells in S-phase simultaneously (Figure 3C). Therefore, we selected miR-127 as the key miRNA in BDNF-induced GC proliferation to continue our research.

BDNF significantly decreased miR-127 expression (Figure 3B) and increased CCND1 expression (Figure 2C). To confirm the regulatory relationship between miR-127 and CCND1, the mRNA expression relationship was determined. As shown in Figure 3C, miR-127 inhibited the expression of CCND1. The protein expression levels of CCND1 were compatible with RT- qPCR after miR-127 was overexpressed or inhibited in porcine GCs (Figure 3D,E).

### 3.4. BDNF Promotes GC Proliferation via the ERK1/2 Signaling Pathway Mediated by miR-127

To determine whether the MAPK-ERK1/2 pathway was entangled in proliferation by porcine GCs after BDNF treatment, the phosphorylation and total protein levels of ERK1/2 in porcine GCs were assessed. The results indicated the phosphorylation levels of ERK1/2 increased after BDNF supplementation for 15 (*p* < 0.05), 30 (*p* < 0.01) and 60 min (*p* < 0.05; Figure 4A). Moreover, GCs were treated with BDNF in combination with PD98059; the effects of BDNF on the viability of GCs (Figure 4B-a) and the distribution of cells in S-phase (Figure 4B-b) were diminished by a blockade of the MAPK-ERK1/2 pathway. Secondly, the expression levels of CCND1 were assessed. Inhibition of ERK1/2 decreased the expression levels of CCND1 mRNAs and proteins (Figure 4B-c–B-e). Thirdly, MAPK-ERK1/2 suppression had no impact on the expression levels of miR-127 (Figure 4B-f), but the phosphorylation of ERK1/2 was stimulated after transfection of miR-127 inhibitor with GCs. Then, after treatment with BDNF, the expression level of p-ERK1/2 was further increased (Figure 4C).

## 4. Discussion

In ovaries, BDNF is mainly presented in follicular granulosa cells and oocytes [16], while BDNF and its receptor TrkB are mainly expressed in granulosa cells and membrane cells of porcine follicles [42]. In this study, the expression levels of BDNF and trkB in GCs were verified by immunofluorescence (Appendix A). BDNF regulates early follicle development in various mammals and directly affects ovulation, as shown in previous studies [10,17]. BDNF has also been associated with different causes of infertility during in vitro fertilization [12,43,44,45]. Additionally, BDNF regulates the maturation of cytoplasmic and nuclear in porcine oocytes by paracrine and/or autocrine signaling systems and promotes potential embryo growth following in vitro fertilization and somatic cell nuclear transfer [10]. Thus, elucidating the transcriptional mechanisms regulated by BDNF in GCs will be beneficial for improving the IVM of porcine oocytes.

Transfection of specific siRNA interfering with BDNF expression in cells will be beneficial to studies of the role of BDNF in the proliferation of porcine GCs. With the extension of transfection time, the interference efficiency of siRNA was enhanced, but further inhibition of GCs proliferation had no significant effect. These findings suggested that the continuous decreases in the secretion of BDNF in porcine GCs did not inhibit cell proliferation at all times.

miRNAs are small noncoding RNAs. They play a vital role in hormone-induced ovarian development [46,47]. In previous studies, the primary role of miR-127 was described as a tumor inhibitor, involved in a series of cellular processes—for instance proliferation, senescence, migration and invasion [48,49,50]. Moreover, miR-127 mediated the differentiation of mouse embryonic endoderms and promoted placental development [38,51]. However, the potential functions of miR-127 in porcine reproduction remain ambiguous. To our knowledge, this study revealed the expression of miR-127 in porcine GCs for the first time, and demonstrated that miR-127 was involved in porcine granulosa cell proliferation as a negative regulator.

Cell proliferation and differentiation are regulated by the progression of the cell cycle. This progression is controlled by cyclins and Cdks complexes [52]. Cyclin D1 (CCND1), a member of the cyclin family, promotes the transition of cells from G1 phase to S phase by interacting with CDK4 and CDK6 [53,54,55,56,57]. CCND1 is also a key regulator of cell proliferation [58]. Due to the inhibition of miR-127 on the proportion of cells in S-phase, the regulatory relationship between miR-127 and CCND1 was examined. The overexpression and knockdown of miR-127 showed that miR-127 significantly negatively regulated CCND1. It also revealed the regulatory pathway, in which BDNF downregulated miR-127 to promote CCND1 expression during porcine GC cell proliferation.

MAPK-mediated signal transduction is a key factor affecting cell fate processes [59]. As the core module of the MAPK signal cascade, the MAPK-ERK1/2 signaling pathway is highly conservative [60]. The extracellular signals are transmitted into the nucleus by ERK1/2 and trigger modifications in the expression of certain proteins in cells [61]. MAPK-ERK12 is also connected with several cellular activities, comprising differentiation, proliferation, apoptosis, transcription and adhesion [62,63,64,65,66,67]. Hence, it was necessary to examine this intracellular signaling pathway to further substantiate the molecular mechanisms underlying BDNF-induced cell proliferation and increased CCND1 expression in GCs. According to our data, phosphorylation of ERK1/2 was increased by BDNF, and porcine GC proliferation was inhibited by blockades of TrkB or ERK1/2. Thus, we concluded that BDNF-induced cell proliferation also depended on MAPK-ERK1/2 signaling activation and that BDNF promoted GC proliferation by regulating CCND1 through MAPK-ERK1/2 signaling cascade. A study on the promotion of bovine GC proliferation by the BDNF-MAPK-ERK1/2 signaling pathway was previously reported [15]. Notably, however, miRNAs (especially miR-127) were found to be involved in the transduction of the BDNF-MAPK-ERK1/2 signaling pathway in this study. Downregulating miR-127 facilitated the stimulation of the MAPK-ERK1/2 signaling pathway. This result suggested that CCND1 upregulation by BDNF indirectly actuated the MAPK-ERK1/2 pathway by downregulating miR-127. These findings provided novel insights for future studies on the function of BDNF and revealed the molecular mechanisms underlying mammalian follicle development. In addition, these findings could provide a new target for the treatment of follicular dysplasia, such as PCOS.

## 5. Conclusions

In conclusion, we determined that BDNF stimulated GC proliferation by increasing CCND1 expression through miR-127 downregulation and MAPK-ERK1/2 pathway activation (Figure 5). Notably, BDNF stimulated the MAPK-ERK1/2 pathway directly and indirectly by downregulating miR-127. Our current findings provided insights that could aid in efforts to effectively promote porcine oocyte maturation and improve embryo production efficiency in vitro.

## Figures and Tables

**Figure 1 animals-13-01115-f001:**
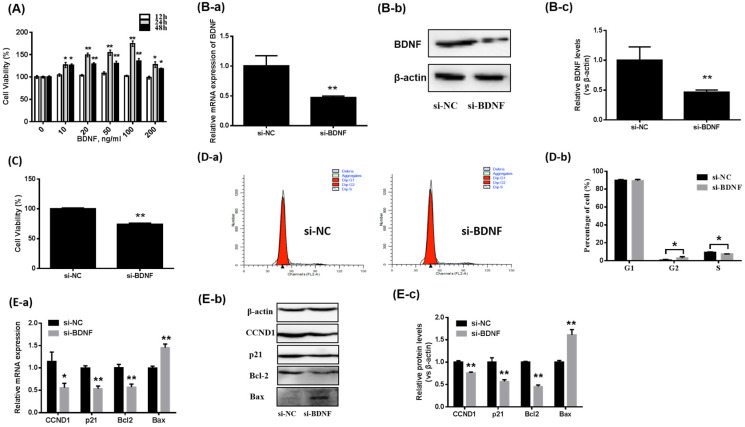
Effects of BDNF on the proliferation of porcine GCs. (**A**) Porcine GCs are cultured with various concentrations of BDNF (0, 10, 20, 50,100, 200 ng/mL) for different times (12, 24, 48 h); CCK-8 assay is performed to examine GC viability. (**B**) Porcine GCs are transfected with si-BDNF or nonspecific siRNA (si-NC) for 24 h. RT-qPCR and Western blotting are performed to assess mRNA (**B-a**) and protein (**B-b**,**B-c**) expression levels of BDNF. (**C**) GC viability is examined by CCK-8 after BDNF knockdown. (**D-a**,**D-b**) The distribution of cell cycle in porcine GCs is determined using FACS. (**E**) The mRNA (**E-a**) and protein (**E-b**,**E-c**) expression levels of Bcl2, Bax, p21 and CCND1 are detected by RT-qPCR and Western blotting after transfection with si-BDNF for 24 h. Data are presented as mean ± SD, *n* = 3; * *p* < 0.05, ** *p* < 0.01. BDNF, brain-derived neurotrophic factor; Bcl-2, B-cell lymphoma-2; Bax, Bcl2-Associated X; p21, cyclin-dependent kinase inhibitor 1A; CCND1, cyclin D1.

**Figure 2 animals-13-01115-f002:**
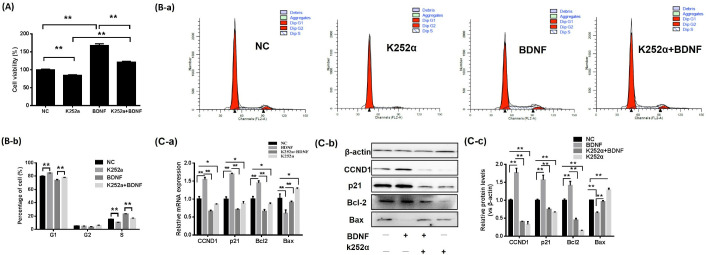
The BDNF/TrkB pathway affects the proliferation of porcine GCs. After 30 min of preincubation with K252a (100 ng/mL) porcine GCs are incubated with BDNF (100 ng/mL) for 24 h. (**A**) CCK-8 assay is then performed to assessed the viability of the GC. (**B-a**,**B-b**) FASC is employed to analyze the cell cycle in porcine GCs. (**C**) RT-qPCR and Western blotting are performed to determine the mRNA (**C-a**) and protein (**C-b**,**C-c**) expression levels of Bcl2, Bax, p21 and CCND1. Data are presented as mean ± SD, *n* = 3; * *p* < 0.05, ** *p* < 0.01.

**Figure 3 animals-13-01115-f003:**
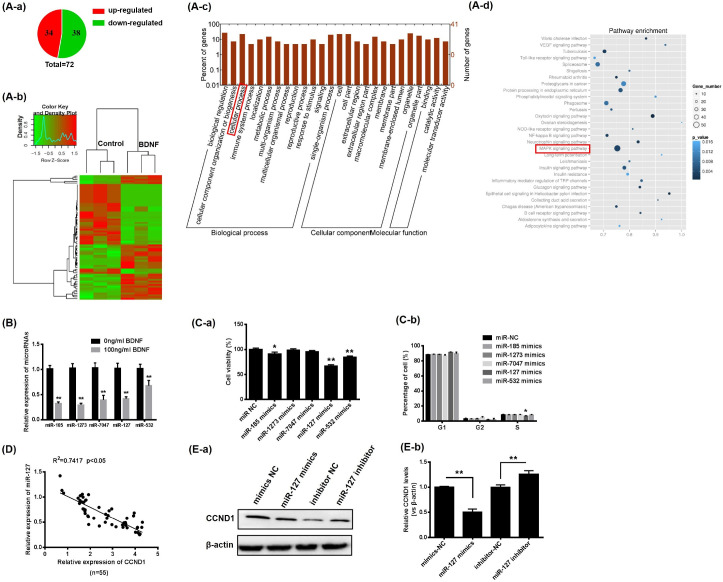
BDNF promotes GC proliferation, increasing CCND1 by downregulating miR-127. (**A**) miRNAs sequencing. (**A-a**) Proportion of upregulated and downregulated miRNAs in GCs. (**A-b**) Hierarchical cluster analysis of the expression pattern of significantly differentially-expressed miRNAs. The highest and lowest fold changes are marked from red to green. (**A-c**) Assessment of miRNA GO enrichment. (**A-d**) miRNA KEGG pathway investigation. (**B**) Relative mRNA expression levels of miR-185, miR-1273, miR-7047, miR-127 and miR-532 in GCs after BDNF (100 ng/mL) treatment for 24 h. (**C**) Transfection of porcine, with miRNAs mimics or negative control, for 24 h. CCK-8 assay performed to investigate the viability of GCs (**C-a**) and FASC performed to analysis the cell cycle (**C-b**). (**D**) Relationship of mRNA expression levels between miR-127 and CCND1. (**E-a**,**E-b**) Porcine GCs transfected with mimics or inhibitors of miR-127 for 24 h. Western blotting is performed to determine the protein expression level of CCND1. Date are presented as mean ± SD, *n* = 3; * *p* < 0.05, ** *p* < 0.01.

**Figure 4 animals-13-01115-f004:**
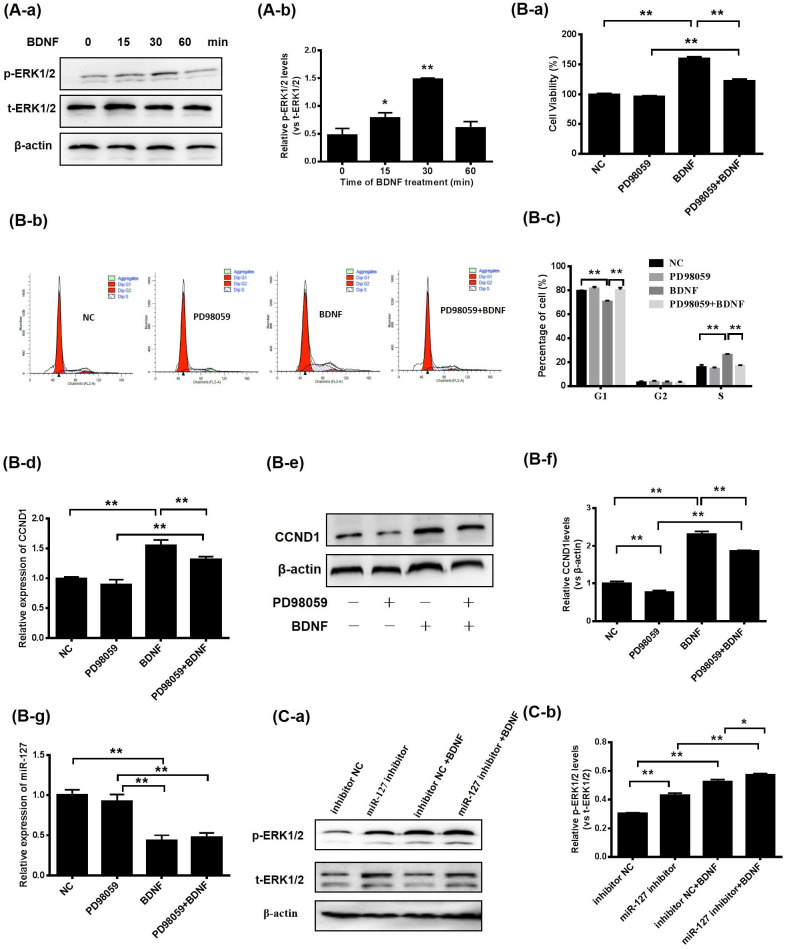
BDNF promotes GC proliferation via the ERK1/2 pathway, mediated by miR-127. (**A-a**,**A-b**) BDNF (100 ng/mL) was applied to porcine GCs for 0, 15, 30 and 60 min. Western blotting revealed the presence of the phosphorylation (p-ERK1/2) and total protein (t-ERK1/2) of ERK1/2. (**B**) Porcine GCs are pretreated with PD98059 (15 μM) for 30 min, before treatment with BDNF (100 ng/mL) for 24 h. The viability of GCs is then investigated using CCK-8 assay (**B-a**). The cell cycle of GCs is studied using FASC (**B-b**,**B-c**). Relative mRNA expression (**B-d**) and protein expression (**B-e**,**B-f**) levels of CCND1 are determined by RT-qPCR and Western blotting. (**B-g**) RT-qPCR is performed to detect the expression of miR-127. (**C-a**,**C-b**) The GCs are transfected with miR-127 inhibitors and then treated with BDNF; the phosphorylation levels of ERK1/2 are detected by Western blotting. Data are presented as mean ± SD, *n* = 3; * *p* < 0.05, ** *p* < 0.01.

**Figure 5 animals-13-01115-f005:**
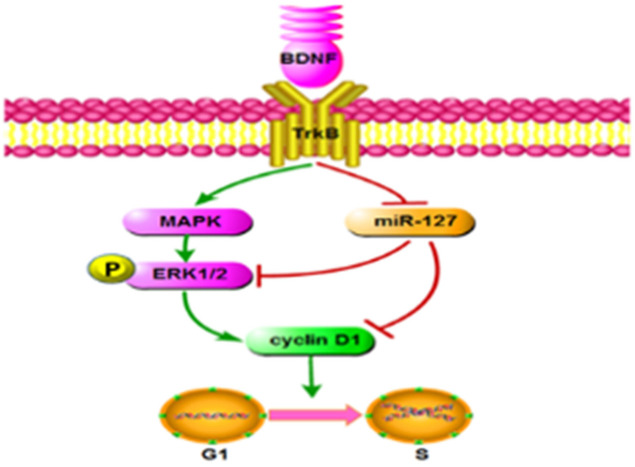
Proposed model of the mechanisms for BDNF, promoting the proliferation of porcine follicular GCs. BDNF directly activates the MAPK-ERK signaling pathway by binding to TrkB and upregulates the expression of CCND1. Subsequently, it promotes the transition of cells from the G1 to S phase. On the other hand, after binding to TrkB, BDNF indirectly promotes the expression of CCND1 by downregulating miR-127.

## Data Availability

All data involved in this article are original and available from the corresponding authors on reasonable request.

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
