# Peer review of "The Mechanisms of BDNF Promoting the Proliferation of Porcine Follicular Granulosa Cells: Role of miR-127 and Involvement of the MAPK-ERK1/2 Pathway"

_animals, 2023, doi:10.3390/ani13061115_

Round 1
Reviewer 1 Report (Previous Reviewer 1)
The authors carefully revised the manuscript in accordance with the reviewer's comments and did a good job in answering all the reviewer's questions.
The manuscript entitled "The mechanisms of BDNF promoting the proliferation of porcine follicular granulosa cells: role of miR-127 and involvement of the MAPK-ERK1/2 pathway" (ID: Animals-2281366) explored the effects of BDNF on the proliferation of porcine follicular granulosa cells in vitro and provided insights for effectively promoting porcine oocyte maturation and improving embryo production efficiency in vitro. The manuscript writing is acceptable and can clearly express the problems found. The research is innovative, and the results can support the conclusion well. Scholars in related fields will be interested in it.
Author Response
Thank you for your comments concerning our manuscript entitled “The mechanisms of BDNF promoting the proliferation of porcine follicular granulosa cells: role of miR-127 and involvement of the MAPK-ERK1/2 pathway” (ID:animals-2281366). All the comments are valuable and helpful for revising and improving our paper. Thank you for your recognition on our research work. We will continue to work hard in the future.
Reviewer 2 Report (New Reviewer)
The manuscript by Zheng et al. examined miR-127 was regulated by BDNF and MAPK- ERK1/2 signaling pathways in porcine follicular granulosa cell (GCs) proliferation in vitro. They found that BDNF promoted porcine follicular GCs proliferation by increasing CCND1 expression by down-regulating miR-127 and stimulating the MAPK-ERK1/2 signaling cascade.
The major conclusions of this research are justified by the results. The methodology seems to be correct in most experiments. However, the study requires improvement in some aspects. Please consider the following points:
Major revision
1. In the part of Introduction, the description of introduction was not enough. The study progress of BDNF in granulosa cells of huaman and animal should be introduced. The role of miRNAs in the productive system should be described, especially miRNA-127.
2. In Figure 1, the results of 3.1 Granulosa cell identification should be described in the supplemental file. Because the identification of granulosa cells is a normal result, there is no innovation. The cellular immunofluorescence of TrkB and BDNF also should be added in the supplemental file, because the expressions of TrkB and BDNF have been reported by other researchers (Wang W S, Cheng L, Zhao X, et al.MiR-15a inhibits the levels of porcine ovarian granulosa cell’s BDNF [J]. Animal Husbandry & Veterinary Medicine, 2018, 50 (3): 14-19).
3. In the part of 3.2 Effects of BDNF on the proliferation of porcine GCs, previous study have demonstrated that BDNF enhances the proliferation of bovine granulosa cells in your own lab, can you explain the difference, meaning, or innovation of your results in Discussion.
4. In Figure 4E-a and 4E-b, the band of β-actin is not clear in Figure 4E-a compare to Figure 5B-e, the analysis of the date seems wrong about miRNA-127 mimics and inhibitor NC. The date should be re-analyzed in Figure 4E-b, Figure 5A-b, Figure 5B-f, and Figure 5C-b.
5. Many of the figures in this manuscript are not clear, such as Figure 3B, Figure 4A-b, A-c, and A-d, and Figure5B.
Minor revision:
1. In line 196, porcine GCs proliferation should be replaced by porcine GC proliferation
2. In line 1390, there is a blank space between CCND1and expression.
Author Response
Please see the attachment

Reviewer 3 Report (New Reviewer)
In this study, Zheng and colleagues investigated the role of miR-127 and involvement of the MAPK-ERK1/2 pathway in BDNF-induced mechanisms, which promote the proliferation of porcine follicular granulosa cells:. On the basis of the results obtained, the authors report that BDNF promoted porcine GCs proliferation by increasing CCND1 expression by down-regulating miR-127 and stimulating the MAPK-ERK1/2 signaling cascade. These findings would be of general interest to this field of research. I have only minor comments.
- The Authors should better discuss in general the regulation of BDNF exerted by miRNA. In this respect I suggest to discuss and add the following studies in the introduction (PMID: 26456533; PMID: 22194877; PMID: 34068160).
- The Authors might want to better discuss about the possible role of other neurotrophic factors in the proliferation of porcine follicular granulosa cells (PMID: 36032306).
- The Authors should also include some statements about the therapeutic implications that might arise from this study
Author Response
Please see the attachment

This manuscript is a resubmission of an earlier submission. The following is a list of the peer review reports and author responses from that submission.
Round 1
Reviewer 1 Report
The manuscript entitled "The mechanisms of BDNF promoting the proliferation of porcine follicular granulosa cells: role of miR-127 and involvement of the MAPK-ERK1/2 pathway" (ID: Animals-2152028) explored the effects of BDNF on the proliferation of porcine follicular granulosa cells in vitro and provided insights for effectively promoting porcine oocyte maturation and improving embryo production efficiency in vitro. The research is innovative, and the results can support the conclusion well. However, for the benefit of the reader, the manuscript still needs some minor modifications before being accepted.
Detailed comments are as follow:
1. Add necessary information in the abstract, such as experimental grouping, treatment concentration, etc.
2. It is recommended not to use first person as much as possible in the manuscript.
3. Line 39 to 41: In conclusion, BDNF promoted porcine GC proliferation by increasing CCND1 expression by downregulating miR-127 and activating the MAPK-ERK1/2 signaling pathway. Delete the second “by”.
4. Line 86: After how many days of culture immune staining was undertaken? Were the cells used for different assays originated from same culture or different cultures?
5. In Figure 1, the Scale bar seems to be blurry and hard to see.
Line 386 to 388: Change “On the other hand, after binding to TrkB, BDNF activates the MAPK-ERK signaling pathway and indirectly promotes the expression of CCND1 by downregulating miR-127.” to “On the other hand, after binding to TrkB, BDNF indirectly promotes the expression of CCND1 by downregulating miR-127.”
6. There are some formatting problems in the manuscript,it is recommended to proofread the manuscript carefully.
(1) Change the format of symbols, such as Line 81, 82, 83, 102, 104, etc.
(2) There are Spaces before and after symbols, such as P < 0.05, etc.
(3) Line 32 to33: “The differential miRNA expression profiles of BDNF-treated group the control group were obtained by miRNA sequencing” Is “The differential miRNA expression profiles of BDNF-treated group and the control group were obtained by miRNA sequencing.” right?
(4) Line 72: change “After” to “after”.
(5) Identify and unify abbreviations for B-cell lymphoma-2: Line 228/Line 243/Figure 2/Figure 3: Bcl2, Bcl-2 or BCL2 ?
(6) Unified time writing format: Line 277: 48h, but the previous expressions were hours. Line 308/Line 321: min, but the previous expressions were minutes.
(7) Check the writing of the concentration unit: Line 223/Line 328
Reviewer 2 Report
This manuscript addresses molecular mechanism that promote the proliferation of porcine follicular granulosa cells. Overall the manuscript is well organized and results are discussed properly . However, some isseus need to be clarified / revised. ,
M&M
· L83-84: Granulosa cells spontaneously luteinize in vitro, so if the cells were cultured for 24h in the presence of 10% FBS before any treatment, are they really GCs or rather luteinized GCs?
· More details are needed on the transfection procedure, e.g. transfection reagent, media, confluency of the cells, and if available, specific sequences for siRNA, miR-mimics and miR-inhibitors.
· Why the authors used such a high dose (50 uM) of miR-mimics and miR-inhibitors? According to Genepharma, the recommended stock solution should be 20 uM, and this should be further diluted during cell transfection.
· Did the authors use a phosphatase inhibitor cocktail during cell lysis for Western blotting? This should be specified.
· More details of antibodies used for Western blotting and IF are needed, e.g. species specificity, source, catalog no.
· Any positive controls for proliferation and cell cycle assays?
· There is some discrepancy between “2.11 Data analysis” and “Figure legends” regarding the expression of data. Is it SEM or SD? Please correct it.
Results:
· Please provide all unprocessed original images of western blots along with their quantification.
· L220-224: the effect of BDNF on GC proliferation was superficially described; what I see in Fig. 2A is that BDNF enhanced cell viability in a dose- and time-dependent manner, and a significant increase was detected, not only for 100 ng/ml but also for other concentrations tested. And that 100 ng/ml was most effective after 24h, which explains why this dose was used for further subsequent experiments.
· L225: what was the transfection efficiency for BDNF mRNA and protein in percent? I have some concern regarding the BDNF protein levels after si-BDNF transfection; although siRNA knockdown is transient, and does not completely reduce the expression of target protein, I would expect more than 50 % reduction.
· Fig. 2D: what statistical analysis was performed here? I suppose t-test was used, but in my opinion it should be two-way ANOVA. Moreover, flow cytometry histogram showing DNA content distribution meaning the % of cells in G1, S, and G2/M is missing, and must be provided. In Line 226 provide the % of siNC and BDNF-treated cells in S-phase. Same comments for Fig. 3, 4 and 5.
· Legend for Fig.2B: Please explain for which assay cells were transfected for 48h. I don’t see this information in M&M section. Moreover (B-c) is not mentioned in the legend.
· K252a (Line 105) or K252α (Line 249) or k252α (Fig.3)? Decide which form you want to use and be consistent throughout the manuscript.
· Fig.3A: Why there is no significant difference between control vs. BDNF? As I mentioned above two-way ANOVA should be performed.
· L277: Here you wrote that cells were transfected for 48h, but in M&M section and in Fig.4 legend you wrote 24h; please correct it.
· I would like to see the efficiency of cell transfection with each mimic and inhibitor. This must be provided and described in the manuscript.
Round 2
Reviewer 2 Report
1. If the authors performed a two-way ANOVA in response to my comments, why wasn’t this mentioned in the “Data analysis” section?
2. I understand the difference between CCK-8 and flow cytometry assays very well, but I don’t understand why the authors say “that positive control is not necessary for proliferation and cell cycle assays.” In accordance with good laboratory practice, it is extremely important to add relevant (positive or negative) control for the in vitro method to validate the executed experiments. There are a number of commonly known controls for proliferation assays that could be used.
3. Regarding si-BDNF transfection, based on my lab experience and general view of gene knockdown, I think an efficiency below 50% indicates that further optimization is needed. Therefore, I cannot agree with the authors that by reducing BDNF mRNA and protein by 34 and 40%, respectively they achieved a good silencing. If the authors had mentioned that they tested different conditions, but these cells are difficult to transfect, which in fact is characteristic for primary cells, I would have no comment. But the authors completely ignore this fact, and in my opinion it should be discussed in the manuscript and supported by other reports.
4. In “Data analysis” I still see that “All data were expressed as mean ± SEM”
5. There is some mistake in the Supplementary Figure 1, the last graph should be dedicated to miR-127 inhibitor and not mimic, right?
6. There are still many typos in the text.

Round 3
Reviewer 2 Report
Despite the fact that the authors have made some changes to the text, there are still some serious issues with the manuscript. The comments in the previous review regarding these issues were generally ignored or only superficially discussed.
The most of the results are of great concern regarding their analyses and interpretation:
1) Western blot analysis
· The raw images for ERK (Figs. 5A-a, 5C-a) include blots for n=3 with extremely different quality, so I would expect to see this ‘variation’ as a much higher SD
· The poor quality of some blots (e.g. for Bax, P21, ERK) and the quantification of the bands that merge between wells, e.g. for B-actin is incorrect; the fact that the authors used Image J software for this analysis does not matter
2) Cell cycle analysis
· I asked the authors to perform a two-way ANOVA and provide a flow cytometry histogram showing DNA content distribution, but this was not done.
· I cannot accept new figure showing only S-phase cells; cell cycle FACS data reveals distribution of cells in three major phases of the cycle (G1, S, and G2) and these are integral part of the analysis and should not be overlooked